# Clinical Applications of Low-Intensity Pulsed Ultrasound and Its Underlying Mechanisms in Dentistry

**Yuzi Wei [1] and Yongwen Guo [2,*]**

[1] State Key Laboratory of Oral Diseases & National Clinical Research Center for Oral Diseases, West China Hospital of Stomatology, Sichuan University, Chengdu 610041, China

[2] State Key Laboratory of Oral Diseases & National Clinical Research Center for Oral Diseases, Department of Orthodontics, West China Hospital of Stomatology, Sichuan University, Chengdu 610041, China

[*] Correspondence: guoyw@scu.edu.cn

**Abstract:** Low-intensity pulsed ultrasound (LIPUS) serves as a non-invasive treatment tool that reaches the lesion site in the form of ultrasound. Due to its low toxicity, low thermal effect, and low immunogenicity, LIPUS has attracted wide interest in disease treatment. It has been demonstrated that LIPUS can activate multiple signal pathways in the shape of sound wave and one of the most acknowledged downstream response components is integrin/focal adhesion kinase (FAK) complex. In recent years, the functions of LIPUS in bone regeneration, bone healing, bone mass maintenance, and cellular metabolism were found. Various oral diseases and their treatments mainly involve hard/soft tissue regeneration and reconstruction, including periodontitis, orthodontic tooth movement (OTM), dental implant, mandibular deficiency, and dentin-pulp complex injury. Thus, more and more researchers pay close attention to the application prospects of LIPUS in stomatology. We searched these articles in PubMed with keywords LIPUS, temporomandibular joint (TMJ), periodontitis, orthodontics, and pulp, then classified the retrieved literature in the past five years by disease type. In this review, the function effects and possible mechanisms of LIPUS in periodontal tissue regeneration, orthodontic treatment, implant osseointegration, TMJ bone formation/cartilage protection, and dentin-pulp complex repair after injury will be summarized. The challenges LIPUS faced and the research limitations of LIPUS will also be elucidated. Therefore, this paper intends to provide new insights into oral disease treatments, explore the optimal application specification of LIPUS, and probe the future research orientation and the prospect of LIPUS in the dental field.

**Keywords:** LIPUS; periodontal regeneration; orthodontics; implants; TMJ; dentin-pulp complex

## 1. Introduction

Low-intensity pulsed ultrasound (LIPUS) is a non-invasive acoustic radiation generally acknowledge to play a significant role in osteogenesis [1]. The intensity of LIPUS ranges from 30 mW/cm$^2$ to 100 mW/cm$^2$. LIPUS shows mechanical vibrations at frequencies above the detection limit of human sound [2]. For avoiding tissue overheating, frequencies of 1~3 MHz and intensity of 30~90 mW/cm$^2$ for 5~30 min/day were often adopted in previous studies [3]. Interestingly, LIPUS appears to have diverse optimal parameters in different tissues and there is still no unified usage specification of LIPUS at present. According to the low-intensity and pulsed pattern, LIPUS not only has the lowest heating effect, but also contributes to transmitting sound waves into specific areas [4]. The pressure waves can serve as mechanical stimulation and activate downstream signals through signal transduction [5]. At present, it's universally acknowledged that the signal transduction is stimulated due to the influence of mechanical stress and/or fluid micro-streaming on the cellular plasma membrane, focal adhesion, and cytoskeletal structures [6,7]. Integrin was found able to serve as the mechanoreceptor on the cell membrane and conduct mechanical

stimulation into cells [8]. When integrin was stimulated by LIPUS signals, the adhesions of focal adhesion adaptor proteins were enhanced. Then, the focal adhesion kinase (FAK) was phosphorylated and promoted signal transduction [9]. After that, the mechanical signal was transformed into electrical or biological signals and then contributed to various biological processes, such as osteogenic differentiation, bone healing, cytokine secretion, and angiogenesis [10–12]. Besides, LIPUS has been implicated to reduce the inflammatory response and mediate tissue regeneration through various signal pathways (Figure 1).

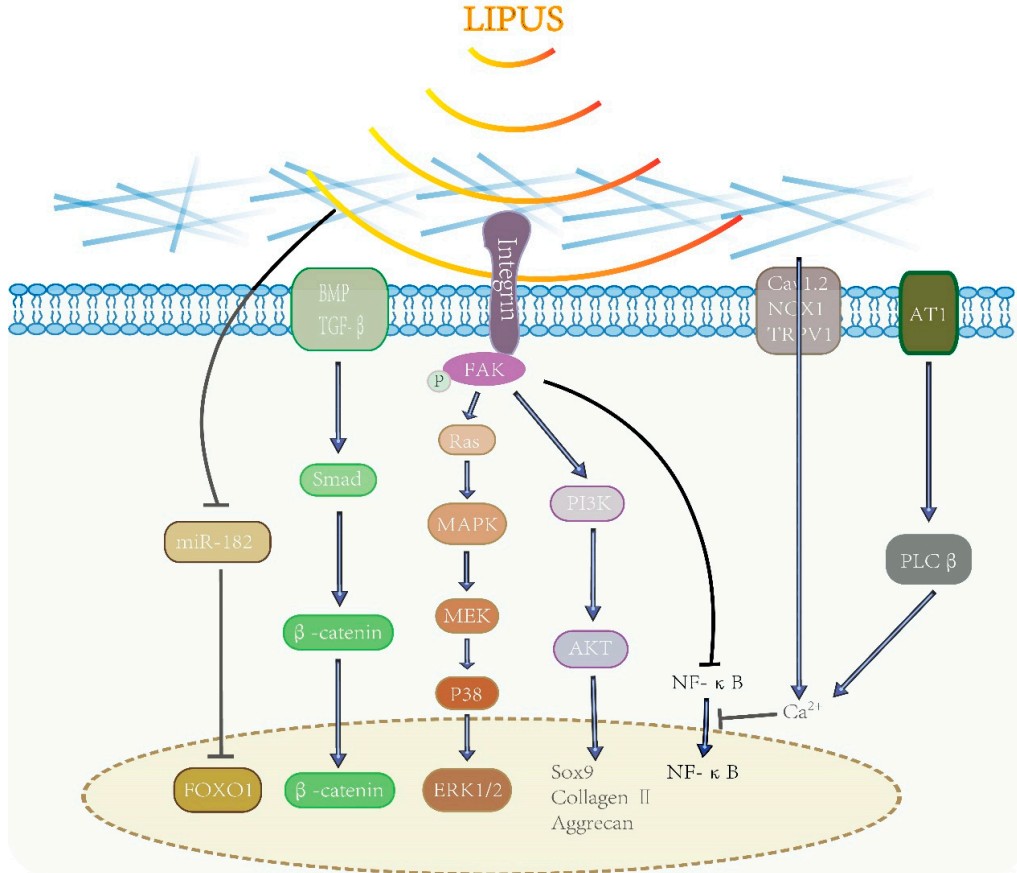

**Figure 1.** LIPUS alleviates oral diseases by modulating multiple signaling pathways. LIPUS can regulate diverse signaling pathways. Integrin and FAK mainly participate in the signal transduction of LIPUS. Through the integrin/FAK complex, the p38 MASK pathway and the PI3K-AKT pathway can be activated to promote osteogenic differentiation. LIPUS also inhibits NF-κB directly. Besides, LIPUS contributes to the AT1-PLCβ-Ca$^{2+}$ pathway and calcium influx. And the increased Ca$^{2+}$ further suppresses the nuclear translocation of NF-κB which promotes the reduction of inflammatory response. LIPUS also accelerates FOXO1 transcription by inhibiting miR-182 and then activates osteogenesis-related factors. Interestingly, Smad has a double effect. On one hand, Smad, which is assembled by LIPUS-activated BMP, plays a role in osteogenesis. On the other hand, LIPUS can stimulate the TGF-β-Smad pathway to repair the dentin-pulp complex.

As for dentistry, oral diseases have influenced a massive number of humans worldwide. In recent years, LIPUS had emerged as a burgeoning therapy in oral diseases and made great advances. The applications of LIPUS in the dental field mainly concentrate on periodontal regeneration, alveolar bone healing, orthodontic bone remodeling, temporomandibular joint cartilage regeneration, and dentin-pulp repair [13–15] (Figure 2). In this review, we integrally elaborated on the clinical application progress and the specific mechanisms of LIPUS in dentistry and expound on the application prospect of LIPUS. In fact, a review concerning recent advances of LIPUS in dentistry for the past five years is lacking. Thus, this summary may bring new insights into the current research state

of LIPUS. Furthermore, we systematically concluded the targets, parameters, and effects of LIPUS in various oral diseases. The limitations and study defects of LIPUS were also set forth. Overall, this paper aims to provide direction for future research on LIPUS in dentistry, find the usage specification of LIPUS to apply this tool more securely and more effectively, and broaden the treatment thoughts for oral diseases.

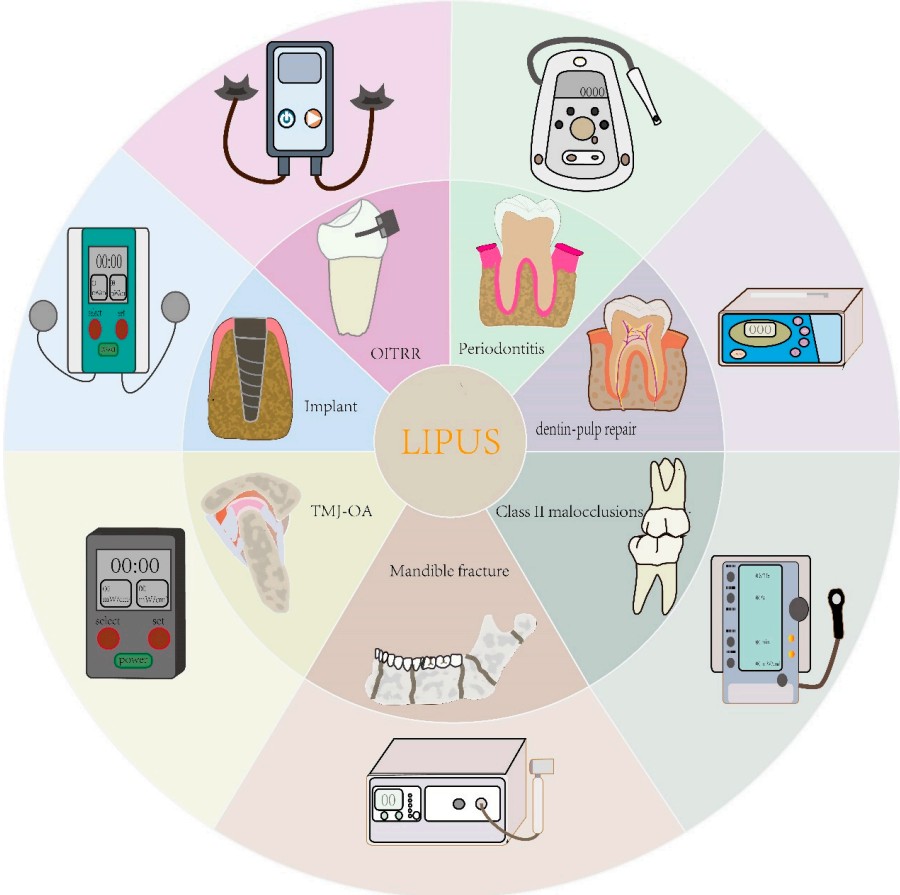

**Figure 2.** The general devices of LIPUS apply to different oral diseases.

## 2. LIPUS Promotes Periodontal Regeneration via Increasing Osteogenic Differentiation and Inhibiting the Inflammatory Response

As one of the most common oral diseases, periodontitis is a kind of devastating disease and relative to microorganisms which is one of the barriers to periodontal regeneration [16]. During periodontitis, broken periodontal tissues (alveolar bone, gingiva, and periodontium), progressive inflammatory reactions, and even tooth loss can be observed, which would significantly influence the supporting function of the periodontal structures, and bring a great deal of discomfort to patients. The conventional treatments for periodontitis include debridement, anti-inflammatory therapy, and allotriodontia [16]. However, the effect of anti-inflammatory is unsatisfactory and traditional tissue regeneration therapy is time-consuming [15]. Based on these existing limitations, it's necessary to discover a new therapy that not only has an influence on anti-inflammatory, but also contributes to periodontal tissue regeneration.

Studies have proven that LIPUS has broad prospects in periodontal diseases. During periodontal wound healing, periodontal ligament cells, osteoblasts of mandibular, and gingival epithelial cells can respond to LIPUS. At the same time, through applying LIPUS, the expression of heat shock protein 70 (HSP 70) was increase, which contributes to wound healing [4]. Presently, LIPUS is considered to have huge potency in periodontal tissue regeneration. Periodontal tissue regeneration can be divided into alveolar bone regeneration,

gingiva regeneration, and periodontium regeneration. It has been demonstrated that LIPUS can promote alveolar bone regeneration, suggesting that it can accelerate periodontal tissue repair [15]. However, several studies found that LIPUS's capacity of promoting periodontal tissue regeneration was mainly relative to the osteogenic differentiation of various cells, including osteogenic differentiations of human periodontal ligament stem cells (hPDLSCs), human periodontal ligament cells (hPDLCs), fibroblasts, and bone marrow stromal cells (BMSCs) [17–19].

HPDLSCs participate in periodontal tissue regeneration owing to their ability to differentiate into osteoblast, mechanocyte, and cementoblast [20]. Moreover, it has been proven that PDLSCs can construct a cementum-periodontium-like composite structure in rats [21]. The application of LIPUS was able to enhance the osteogenic differentiation of hPDLSCs and promote the regeneration of periodontal ligament (PDL) like tissue [14]. Through using LIPUS, more extracellular matrix (ECM)-related genes were found, including constans-like (COL-1, COL-3), fibronectin [22], integrinβ-1, and laminin (LN). These genes contributed to cell differentiation, cell growth, and cell proliferation. Specially, COL-1 participated in forming PDL, which is also called periodontium [23–25]. The increasing genes can promote the synthesis of ECM and the growing ECM enhanced the functions of hPDLSCs [14]. Besides, another study supposed that LIPUS played an important role in inhibiting endoplasmic reticulum stress (ERS) and then reducing unfolded protein response (UPR), and finally facilitating PDLSCs osteogenic differentiation [17]. The nuclear translocation of NF-κB can inhibit PDLSCs osteogenic differentiation. LIPUS is able to inhibit the NF-κB pathway to block nuclear translocation [26]. In addition to osteogenic differentiation, LIPUS also has an influence on the migration of PDLSCs. LIPUS may promote PDLSCs through the stromal cell-derived factor-1 (SDF-1)/C-X-C motif chemokine receptor 4 (CXCR4) signaling pathway. In this process, twist family bHLH transcription factor 1 (TWIST1) serves as the upstream factor of SDF-1 [27–29]. Moreover, the migrated PDLSCs contributed to PDL regeneration and repair [30].

PDLCs could express bone-associated molecules and stimulate growth factors [31]. These expressed factors equip PDLCs with osteogenic differentiation capacity and play an important role in the regeneration and repair of periodontal tissue [32]. It was demonstrated that LIPUS had an anabolic effect on PDLCs and increased their pluripotent characteristics [18]. Multiple signaling pathways take part in the osteogenic differentiation of PDLCs induced by LIPUS. A research team found that the p38 MASK pathway, integrin β1-dependent signaling transduction, and Runt-related transcription factor 2 (Runx2) were all concerned with the osteogenic differentiation of PDLCs [33,34]. By inhibiting the p38 MASK pathway and integrin β1, the expression of bone-associated factors was decreased, e.g., alkaline phosphatase (ALP), Runx2, and osteocalcin [33,34]. Bone morphogenetic protein (BMP) can induce bone-associated factors including Runx2 and β-catenin to gather to the osteocalcin promoter [35]. When Smad serves as an effector of BMP, the osteogenic differentiation of PDLCs can be greatly promoted [36]. In addition to positive regulation, LIPUS also plays a negative regulating role. After post-transcriptional regulation, miR-182 was able to act as the inhibitor of transcription factor Forkhead box O1 (FOXO1) and then reduce the activation of ALP and Runx2 [37,38]. LIPUS was shown to restrain miR-182 expression [39].

Since periodontal ligament fibroblasts (PDLFs) were also able to express bone-associated proteins, they acted as the target of growth and differentiation factors [40]. Further, in periodontal tissue metabolism, PDLFs can serve as either osteoprogenitor cells or immune-modulatory cells [41]. LIPUS has two types of adjustment to the osteogenic differentiation of PDLFs. On one hand, through the ROCK signaling pathway, LIPUS reduced the inhibiting effect of lipopolysaccharide (LPS), interleukin (IL-1β), and tumor necrosis factor (TNFα) on PDLFs osteogenic differentiation [10]. On the other hand, the stimulation of LIPUS increased the incentive function of BMP-9 to PDLFs osteogenic differentiation [10,42]. In addition to PDLFs, LIPUS can enhance the potency of human gingival fibroblasts (HGF) osteogenic differentiation through increasing the activation of ALP and the expression of

osteopenia (OPN) [43]. BMSCs are also able to differentiate into osteoblasts in a specific condition. Through comparing the BMSCs with LIPUS exposure and without LIPUS exposure, it was proven that BMSCs with LIPUS exposure had more alveolar bone formation, and the expression of COL and OPN also increased. This phenomenon indicated that the combination of LIPUS and BMSCs contributed to periodontal alveolar bone regeneration [19].

In fact, LIPUS also plays a crucial role in controlling the inflammatory response. The periodontal inflammatory response has a negative influence on periodontal tissue regeneration so reducing inflammatory response is necessary [44,45]. The periodontal inflammatory response was mainly caused by LPS, which promoted the expression of various inflammatory cytokines [46]. It was demonstrated that the LIPUS-inhibited NF-κB signaling pathway can reduce the expression of LPS-promoted inflammatory cytokines, including IL-6 and IL-8 [18]. LIPUS can inhibit NF-κB pathway through diverse upstream signals. Angiotensin II receptor type 1 (AT1) can respond to LIPUS and then activate the PLCβ-$Ca^{2+}$ pathway [47]. By using AT1 antagonist Losartan, all the functions of LIPUS to LPS-induce IL-1a and NF-κB nuclear translocation were restrained, while using PLCβ inhibitor U73122 may regain the NF-κB nuclear translocation. This phenomenon suggested that LIPUS inhibits the activation of NF-κB and the expression of LPS-induced IL-1a through the AT1-PLCß pathway [48]. Besides, LIPUS can also reduce LPS directly and inhibit inflammatory signals induced by IL-1β and TNFα. This mechanism was possibly related to ROCK1 and other small G-proteins [10].

In a word, LIPUS promotes periodontal regeneration mainly through increasing osteogenic differentiation and inhibiting inflammatory responses (Figure 3). By further exploration of LIPUS, it's possible to find diverse insights and targets in periodontal regeneration. Many conclusions of present studies are based on vitro studies. According to the discrepancy, more research in vivo needs to be designed and conducted in order to have a more accurate understanding of the functions of LIPUS.

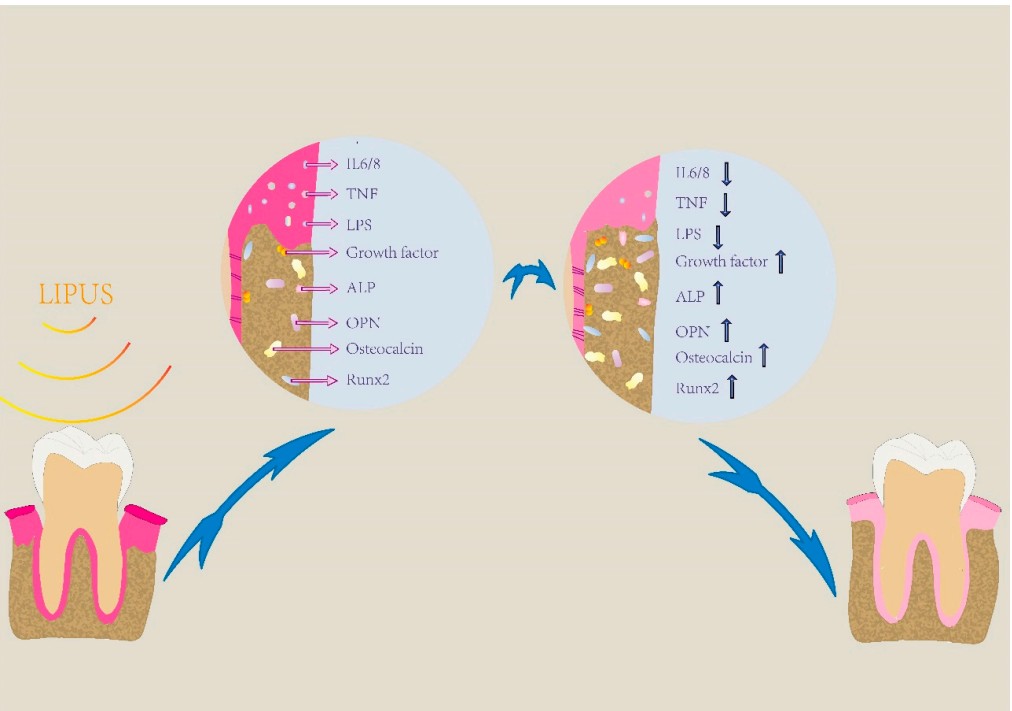

**Figure 3.** LIPUS plays a dual role in periodontitis. By applying LIPUS, inflammatory factors including IL-6, IL-8, TNF decrease while osteogenesis-related factors increase. Therefore, the inflammatory response and alveolar resorption of periodontitis can be significantly improved.

### 3. LIPUS Plays a Significant Role in Orthodontic Treatment via Accelerating OTM and Alleviating OITRR

Malocclusion is a common oral problem which may influence not only oral health and function, but also facial esthetics and even mental health [49]. With the propagation of facial and dental aesthetics, more and more patients choose orthodontic treatment to improve their appearance. However, due to the different speeds of orthodontic tooth movement (OTM), the usual duration time of orthodontic treatment ranges from one year to three years and some patients even have to bear longer periods [50]. Such a long time becomes a scruple for patients and some side effects of orthodontic treatment are more possible to happen, including caries, inflammation, and orthodontically induced tooth root resorption (OITRR) [45,51]. Besides, during orthodontic treatment, patients will experience various degrees of pain which may affect appetite and moods [52]. The above scenarios immensely affect patient compliance and block the progress and effect of orthodontic treatment. Therefore, finding an adjuvant therapy is urgent to improve these problems. LIPUS has become a hopeful candidate in expediting OTM, reducing OITRR [8].

The wires and brackets or clear aligners in teeth can respond to external mechanical forces, and then OTM will happen, which is a complicated process of bone remodeling. During OTM, tooth can be divided into pressure zone and tension zone. The former is in the movement direction of the tooth while the other is opposite [53]. In the pressure side, bone resorption will occur which as a rate-limiting factor to OTM [54]. LIPUS was demonstrated to promote OTM in both human and animal. A randomized controlled trial (RCT) found whether applying LIPUS decreased the treatment time to 49% and the patient compliance increased to about 66% by comparing patients using LIPUS and Invisalign SmartTrack® clear aligners with those only using the aligners [55]. Another RCT also obtained a similar outcome in that the average rate of tooth movement rose 29% by exposure to LIPUS [56]. The above two RCTs all suggested that LIPUS have a vital impact on promoting OTM. RANK, receptor activator of nuclear factor kappa-B ligand (RANKL), osteoprotegerin (OPG), and RUNX-2 are relative to osteogenesis and bone remodeling. In rats, the expression of these factors would increase by applying LIPUS [57]. Actually, in ovariectomized osteoporotic rats, LIPUS was able to maintain normal OTM, which indicated that LIPUS may also have an effect on postmenopausal women [58]. In fact, the mechanism of LIPUS in promoting OTM is still unclear. A study found that after the stimulation of LIPUS, the number of BMP-2 positive cells and the expression of RANKL were obviously rising. The Hepatocyte growth factor (HGF)/Runx2/BMP-2 signaling pathway and RANKL may participate in LIPUS-induced OTM improvement [59] (Figure 4). In fact, low-level laser therapy (LLLT) can promote bone remodeling similarly, so it is also applied in orthodontic treatment [60,61]. While LIPUS has a greater effect on osteoblast, LLLT mainly aims at osteoclast. Interestingly, applying LIPUS, LLLT, LIPUS, and LLLT in mice respectively, the bone remodeling effect was conspicuous in the LIPUS and LLLT group while lowest in the LIPUS group [57]. This outcome suggests that the combination of LIPUS and LLLT is more effective in facilitating OTM.

OITRR usually occurs in orthodontic treatment as a side effect [62]. A study showed that almost all patients and 55%~91% teeth occurred root resorption during orthodontic treatment [63]. However, OITRR has few clinical symptoms and a clinically available approach for treating OITRR has not been discovered. So as OITRR is examined, it's often severe [64]. Under most circumstances, only the surface layers of cementum are assimilated and then repaired by cementoblasts during OTM. Once the repair ability of the cementum was suppressed, apical root resorption becomes worse [65]. LIPUS was proven to impose restrictions on OITRR. Applying LIPUS 20 min/day in beagle dogs that received OTM, cementum in premolars near the apex became thick, indicating the potential of LIPUS in reducing OITRR [66]. In beagle dogs, the LIPUS inhibiting effect of resorption in the middle and the apical thirds of the root was prominent [67]. Similarly, in rats, the number of osteoclasts, resorption lacunae, and resorption area ratio in the group with LIPUS stimulation were all lower than in the group only with orthodontic appliances [68].

Concerning the mechanism of LIPUS inhibiting OTIRR, one of the most acknowledged viewpoints is that OPG and RANKL play an important role. RANKL can combine with its receptor RANK on the surface of osteoclast precursors and then promoted the growths and differentiations of osteoclasts [69]. OPG is an inhibitor of RANKL which can reduce the absorption of alveolar bone and promote the apoptosis of osteoclast [70]. However, studies discovered that LIPUS contributed to bone repair and remodeling through increasing the proportion of OPG/RANKL (Figure 5). The increased OPG/RANKL enhanced the ability to inhibit osteoclasts and then reduce the destruction of cementum, thereby suppressing OITRR [62,68]. In addition to LIPUS, photobiomodulation (PBM) also has an effect on OITRR. Actually, a control experiment manifested that the function of LIPUS and PBM in OITRR had no significant difference [68]. However, at present, research on the combination of LIPUS and PBM in OITRR is limited. More control experiments are needed to explore whether the cooperation of LIPUS and PBM is more efficient in OITRR.

In addition to OTM and OTIRR, LIPUS was demonstrated to play a crucial role in relieving pain. Through applying LIPUS, osphyalgia can be reduced [71]. In the field of the maxillofacial region, LIPUS is also effective. It was proved that LIPUS is an efficient method to relieve pain after orthognathic surgery [72]. Based on these results, LIPUS was tried to use in orthodontic pain. However, through applying LIPUS to patients who have Angle's Class II division 1 malocclusion every three weeks, the orthodontic pain showed no improvement, which indicated that LIPUS had no analgesic effect in these patients [73]. Since there are limited studies about LIPUS and orthodontic pain, further research is needed to explore their relationship.

In general, LIPUS has a vital effect on enhancing OTM and reducing OITRR. Actually, the frequency of LIPUS used in OTM is usually 1.5 MHz. Since different frequencies may generate diverse outcomes, it's necessary to study ulteriorly to explore the best frequency and the usable range of LIPUS. The bright spot of LIPUS is that it can promote the treatment of malocclusion and OITRR at the same time and prevent the further development of OITRR during orthodontic treatment. So, more attempts to design and manufacture a more convenient LIPUS device through interdisciplinary cooperation are urgent.

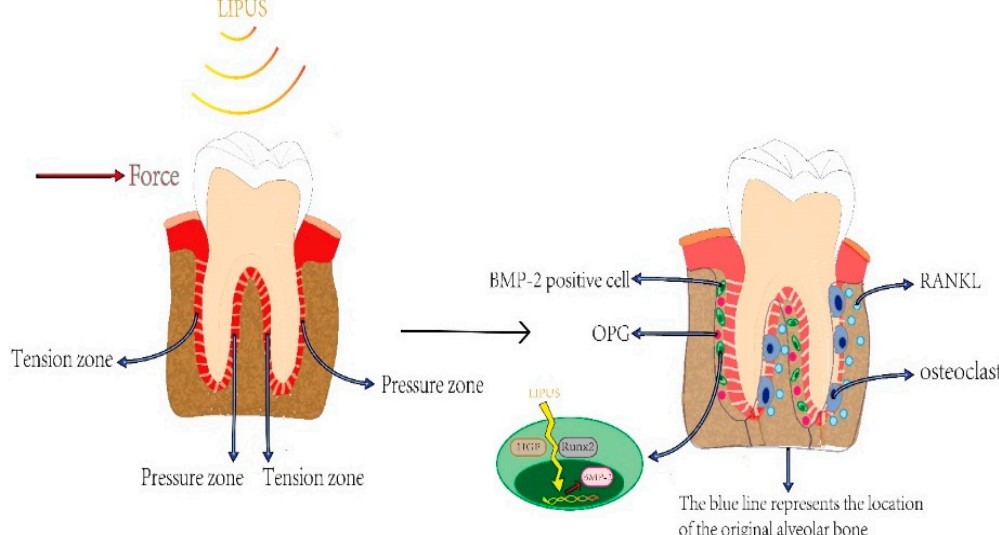

**Figure 4.** LIPUS has an effect on both the tension zone and the pressure zone in the OTM process. During OTM, the tension zone happens to bone remodeling while the pressure zone occurs bone resorption. LIPUS is able to stimulate the expressions of osteoclasts and RANKL in the pressure zone and promote the production of OPG and BMP-2 through the HGF/Runx2/BMP-2 pathway in the tension zone, thereby expediting OTM.

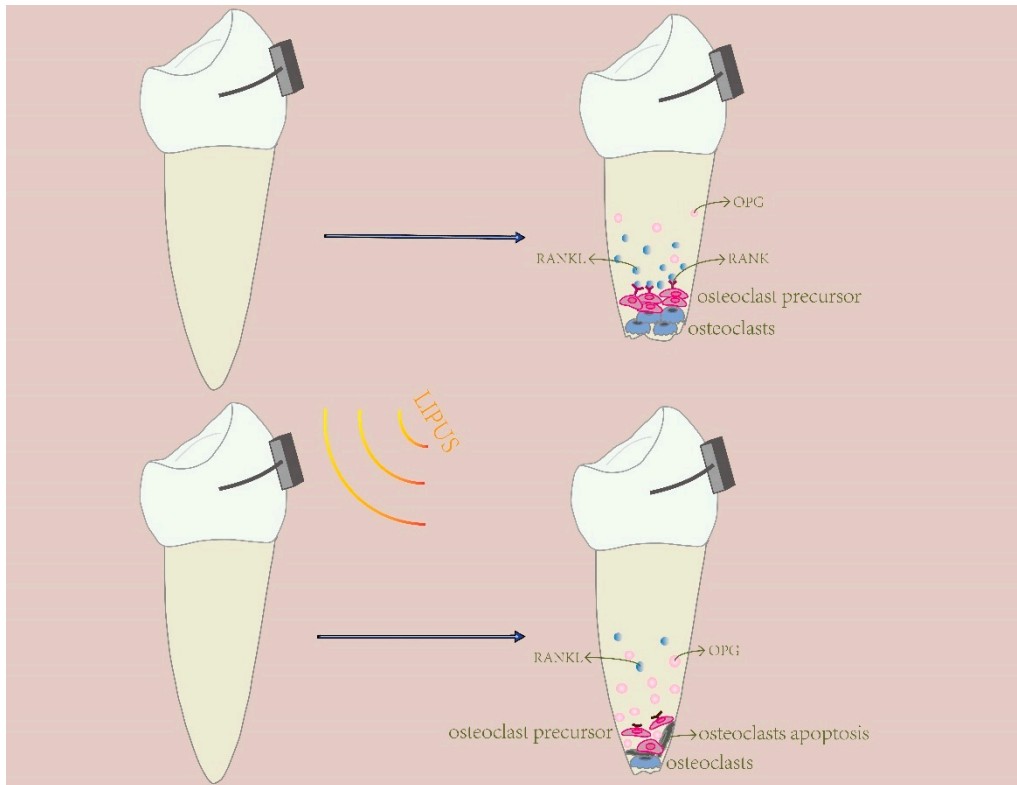

**Figure 5.** LIPUS inhibits further exacerbation of OITRR by affecting the ratio of RANKL and OPG. During OITRR, the ratio of RANKL/OPG is high. The RANKL binds to RANK which exists in the osteoclast precursor and then the osteoclast precursor differentiates into osteoclasts. The growing osteoclasts lead to OITRR further aggravation. Through using LIPUS, the ratio of RANKL/OPG decreases which contributes to osteoclast apoptosis.

## 4. LIPUS Contributes to the Stability of Implants via Facilitating Osseointegration

Dental implants have been widely applied in the mandible and the maxilla. However, a lack of osseointegration may result in implant looseness and the stability of implants depends on the surrounding bone and the bone implant interface (BII) [74]. Less alveolar bone and osteoporosis may also lead to implant failure. Furthermore, building a secure surface between the implant and surrounding bone is a slow process [75]. At present, there are numerous studies exploring new technology to improve and promote osteoporosis of dental implants. Ultrasound techniques had been proven that can stimulate dental implants' osteoporosis and the spread of ultrasonic waves in dental implants can be led by the structure of implants [76,77]. Due to the excellent performance in osseointegration and soft tissue healing, LIPUS attracted broad attention in the context of implants [78]. LIPUS was found to contribute to peri-implant bone volume at any time and promote the contact between bone and implants from the fourth week [79]. Besides, the combined application of LIPUS and low-magnitude high-frequency (LMHF) loading can also enhance the osteogenic activity of peri-implant bone and osseointegration can be observed in about the fourth week after implanting [80]. Actually, in the studies of rabbits, pigs, and rats, LIPUS was also demonstrated as effective to enhance osseointegration.

Since the jaws of rabbits, pigs, and rats are thin, it's difficult to build a model of animal maxillofacial implants. So, most of the studies used dental implants in the tibial model to evaluate the effect of osseointegration. Using LIPUS in a rabbit tibial model with dental implants imbedded, the bone area values, the bone volume values, and the bone-implant contact values were all increased significantly while the increase of bone area values and bone volume values was earlier than that of bone-implant contact volume [81]. Besides, the fibrillar calcified layer and trabecular bone around the dental implant in

rabbits with LIPUS occurring earlier than in rabbits without LIPUS. Further, the maximal pull-out strength and stiffness were more preeminent on the side with LIPUS, which suggested that the combination of dental implants and bone is excellent [75]. Similarly, in the pig model, LIPUS can also promote bone formation and bone-implant contact surface healing [82]. Interestingly, a rat model was built with dental implants in the maxillary first molars extraction sockets successfully. In this study, LIPUS played a significant role in promoting the dental implant osseointegration of αcalcitonin gene-related peptide (αCGRP) positive rats while cutting no ice in αCGRP negative rats. This consequence indicated that LIPUS may promote local neurons to generate αCGRP to enhance dental implant osseointegration [83]. In addition to animal studies, the human study also declared that LIPUS contributed to dental implant osseointegration. Through fractal dimension (FD) analysis, dental implants with LIPUS had better stability and the side which was closer to LIPUS showed more new bone formation [84].

In addition to dental implants, miniscrew implants (MSIs) are also needed in orthodontic treatment. MSIs can be implanted into the alveolar bone to pull tooth movement [85]. However, the success rate of implanting MSIs is about 86.5%, which is lower than dental implants [86]. The surrounding bone metabolism of MSIs is one of reasons for MSI failures [87]. Since LIPUS can promote bone formation, it was applied to enhance the stability of MSIs. Through illuminating LIPUS, not only Ti-6Al-4V MSIs, but also stainless steel (SS) MSIs contacted more closely with the surrounding bone. Besides, the cortical bone density, cortical bone thickness, and cortical bone rate all increased after applying LIPUS [87]. Another controlled experiment observed that the incorporation of MSIs and surrounding bone in the group with LIPUS exposure was stronger than that in the group without LIPUS exposure [88]. These two studies all demonstrated that LIPUS was able to reduce MSIs looseness and promote bone healing in implanting position.

As a matter of fact, due to the limited human experiment, the use of LIPUS in implants is still immature. Moreover, at present, most animal experiments are based on tibia rather than jaws. More animal jaw implant models are necessary to be built and more jaw-based animal experiments need to be expanded. Further human trials are urgent to ensure the feasibility of LIPUS in humans and explore the optimal frequency of LIPUS for human implant osseointegration.

## 5. LIPUS Makes an Effect on TMJ-Involved Diseases via Expediting Bone Formation and Defending Chondrocytes

The temporomandibular joint (TMJ) mainly consists of the mandibular condylar, the articular surface of the temporal bone, and the articular disc [89]. As the only left and right linked joint in the maxillofacial region, the TMJ plays an important role in chewing, speaking, and facial movements. However, on account of the scarce oxygen, the only cells that exist in the condylar tissue of the joint are chondrocytes [90]. Many TMJ-involved diseases are relative to the bone and articular cartilage. LIPUS was proven to contribute to bone healing and the repair of cartilage injury [91,92]. So, applying LIPUS in TMJ-involved disease anesis may be a potential approach.

TMJ osteoarthritis (TMJ-OA) is a usual disease in the maxillofacial region, which may occur cartilage degradation, subchondral bone remodeling, and bone wear [93]. TMJ-OA may result in pain and difficulty in the opening mouth due to the interaction between the generated blood vessel and invading nerves [94]. In this process, chondrocytes in TMJ generally undergo apoptosis and the cartilage matrix is easily degraded [95]. According to a direct mechanical injury and hypoxia/reperfusion injury model, a lack of oxygen in the articular cavity is able to induce the production of oxygen radicals and the oxygen radicals could result in joint injury and inflammatory response [96]. Owing to the ability of tissue regeneration, LIPUS gradually comes into view. LIPUS was demonstrated to not only protect the cartilage, but also reduce the inflammatory response through various approaches. In a simulative oxygen-deficient environment, through applying tandem mass tag (TMT) technology, it was observed that LIPUS increased the expression of chondrogenic

factors, including sox9, collagen II and aggrecan, while reducing the expression of vascular endothelial growth factor (VEGF), which suggested that LIPUS can promote chondrocytes growth and inhibit angiogenesis in hypoxia-Induced TMJ-OA. Besides, the study also found that extracellular Matrix (ECM) proteins may become fresh treatment targets [97]. However, hypoxia contributes to activating the hypoxia-inducible factor pathway, and the expression of hypoxia inducible factor (HIF) existing in chondrocytes increases. Interestingly, LIPUS was found to decrease the expression of HIF-2. This outcome indicated that HIF-2 may be the downstream targets of LIPUS in TMJ-OA [98]. In addition, through RNA sequencing, matrix-degrading enzyme (ADAMTS-8) and circadian gene (Per-2, Dbp, Npas2, and Arntl) were also considered to have a close relationship with TMJ-OA [99,100]. Through LIPUS, the expression of Per-2 increased while the expression of ADAMTS-8 decreased. In a rat TMJ-OA model, $Zn^{2+}$ exporter-9 (ZNT-9) could be activated by LIPUS. The increased ZNT-9 had an effect on down-regulating ADAMTS-8 and up-regulating aggrecan in chondrocytes, which contributed to the protection of chondrocytes [99]. Based on the above results, studies of ZNT-9 or ADAMTS-8 and LIPUS may provide novel insights into the treatment of TMJ-OA. Moreover, in the rabbit TMJ-OA model, LIPUS was proven to inhibit inflammatory cytokine interleukin-6 to anti-inflammation and this inhibiting effect was mediated by the transforming growth factor-β1 (TGF-β1)/Smad3 pathway [101]. Synovial tissue plays a significant role in TMJ nutrition and lubrication. LIPUS can lower the expression of matrix metalloproteinase-9 (MMP-9) which participates in the production of cytokines and tissue injury in inflammation [102]. Moreover, immunohistochemistry indicated that MMP-9 is over-expressed in the synovial tissue of OA patients [103]. Thus, LIPUS may have potency in reducing the inflammation of synovial tissue in TMJ by inhibiting MMP-9 expression. Therefore, LIPUS was considered as a hopeful therapeutic tool of TMJ-OA on account of protecting articular cartilage/chondrocytes, reducing ECM degradation, and alleviating inflammatory reactions.

Mandibular deficiency, which is characterized by mandibular retrusion, is one of the usual reasons for class II malocclusions [104]. Mandibular deficiency influences the appearance and function of TMJ, so it's necessary to stimulate bone formation in the mandible [105]. However, some studies demonstrated that LIPUS was able to accelerate bone formation in condyle. Applying LIPUS in TMJ of rats for 20 min/day, it could be observed that both the thickness of fibrous cartilage and the number of cells increased. Endochondral bone formation and subchondral trabecular bone remodeling in condyle were also promoted obviously. These phenomena suggested that LIPUS contribute to cartilage and bone formation of condyle [106]. Thus, the application of LIPUS in mandibular deficiency is feasible. Besides, Micro-CT analysis showed that the bone volume fraction in rats with LIPUS treated was higher than that in rats without disposal. Interestingly, the bone volume in rats with LIPUS and basic fibroblast growth factor (bFGF) treated was uppermost [107]. This study not only showed that LIPUS can stimulate the growth of mandibular condyle, but also revealed that there is a synergistic effect between LIPUS and bFGF. Similarly, through histological examination and Micro-CT, it was demonstrated that the cooperation of LIPUS and functional appliances had a positive influence on condyle growth and chondrocytes proliferation [108]. However, the usage time of LIPUS can also affect the effect of condyle growth. When TMJ was irradiated with LIPUS for 10 min/day, 20 min/day, and 40 min/day, the greatest increase in trabecular perimeter was observed in rats with LIPUS treated at 20 min per day, implying the best effect on condylar growth [109,110]. Taken together, LIPUS can improve mandibular deficiency, especially the growth of the condyle, and it works best when used for 20 min/day.

Mandible fracture is the second most common fracture in the maxillofacial region [111]. It had been demonstrated that the cells in mandibular fracture hematomas had the potency of osteogenesis [112]. In an animal experiment, LIPUS had been found that had the prospect in mandible fracture treatment. In the rabbit unilateral mandibular interceptive model, after the application of LIPUS, the rigidity of healing bone became stronger, meaning greater mechanical property. Besides, according to histomorphometric analysis, both

bone volume and bone densities all remarkably increased in LIPUS-treated rabbits [113]. Mandibular fracture hematoma-derived cells (MHCs) had the capacity for osteogenic differentiation and may contribute to bone healing in mandibular fracture. Increased expression of osteogenic-related factors, such as ALP, Runx2, and OPN, can be observed in human MHCs irradiated with LIPUS. The result suggested that LIPUS can promote MHCs osteogenic differentiation and LIPUS has shown potential to be applied in bone healing [114]. Fortunately, previous studies explored the mechanism of LIPUS regulating MHCs expression. In MHCs, LIPUS was able to stimulate the expression of BMP, which has an effect on inducing bone formation. Thus, BMP may participate in the stimulating effect of LIPUS on MHCs [115]. The proposed mechanism may provide new ideas and new targets for the treatment of mandibular fractures. As a matter of fact, the effect of LIPUS on mandibular fractures was demonstrated in RCTs in humans. A total of 68 samples were included in two RCTs. Mandibular fracture healing was found to be greater in the experimental group than in the control group, as assessed by ultrasound. In addition, the pain index was lower in the study group than that in the controls, as evaluated by the patients themselves [116,117]. Therefore, LIPUS played a double role in a mandibular fracture, and it can not only accelerate bone healing, but also reduce pain.

In fact, LIPUS can also play a role in temporomandibular joint disorders and temporomandibular joint injuries. At present, great progress has been made in the role of LIPUS in the TMJ. LIPUS not only has a protective effect on articular cartilage, but also contributes to bone healing and maintenance of bone mass. In addition, in TMJ-related diseases, LIPUS has been proven to be effective in the human body to alleviate symptoms, supporting its broad application prospects.

## 6. LIPUS Repairs the Dentin-Pulp Complex via Stimulating Tertiary Dentin Formation

The formation of a dental-pulp complex is always based on severe caries stimulation and the complex plays a reproducible, protective, supportive, and nutritive role [118]. Dental can be divided into primary dentin, physiologic secondary dentin, and tertiary dentin. Especially, tertiary dentin is formed in the corresponding position within the dental pulp cavity, which is a defense response to external injury stimuli [119]. Therefore, the formation of tertiary dentin has great significance in the repair of the dentin-pulp complex after injury. LIPUS was shown to promote tertiary dentin formation and then protect the dental pulp.

A mouse model of dentin-pulp complex injury was constructed by creating a carious cavity in the maxillary first molar of mice. The use of LIPUS made the tertiary dentin accumulate continuously, and the inflammation of dental pulp tissue decreased [120]. The regulatory mechanism of LIPUS on the dentin-pulp complex was reported. TGF-β1 can stimulate odontoblast differentiation of dental pulp cells, and Smad is involved in its signal transduction [121]. The expression of TGF-β1 and Smad increased after dentin injury in the maxillary first molar of mice. However, through the application of LIPUS, the expressions became higher, which suggests that LIPUS may promote dentin-pulp complex repair by regulating the TGF-β1/Smad signal transduction pathway [122]. In addition, another study found that LIPUS can enhance the expression of calcium ion transport-related proteins (Cav1.2, NCX1, and TRPV1) and stimulate tertiary dentin formation [123]. In the tooth slice organ culture experience, the number of odontoblasts increased after irradiation with LIPUS, also indicating the feasibility of repairing the dentin-pulp complex with LIPUS [124,125].

The dentin-pulp complex can be influenced by orthodontic treatment. During orthodontic treatment, the pulp cavity may decrease and the organization structure of pulp may change [126]. The dental pulp was able to react to orthodontic forces. After the orthodontic force invaded, the cell layer of the pulp was damaged, and the blood vessels of the pulp were congested and dilated. More seriously, the pulp was infiltrated by a large number of chronic inflammatory cells, leading to inflammation [127]. On one hand, LIPUS plays a significant role in the repair of dentin-pulp complex injury. On the other hand,

LIPUS can shorten orthodontic treatment time and then reduce the invasion of orthodontic force. Therefore, LIPUS may have a protective effect in dental pulp injury induced by orthodontic force and may be a potential application tool to mitigate dental pulp injury.

In fact, there are limited studies on the application of LIPUS in the dentin-pulp complex. LIPUS may also be harmful to roots because the increase of tertiary dentin may make the root canal narrow. In addition, current studies focus on animals or in vitro, so more studies based on the human body are necessary to determine the true significance of LIPUS to the dentin-pulp complex.

### 7. Conclusions and Prospect

As a non-invasive treatment, LIPUS has been found to contribute to bone regeneration, bone remodeling, and cell metabolism. It has been considered as a novel therapeutic tool in dentistry. The feasibility of LIPUS has been proven by both animal experiments and human control experiments. As previously described, LIPUS has great potential in periodontal tissue regeneration, OTM-related periodontal remodeling, implant bone integration, TMJ bone formation, and dentin-pulp complex repair. We discovered that the TGF-β1/Smad pathway has an effect on both TMJ-OA treatment and dentin-pulp complex repair, so this pathway may be a therapeutic target for TMJ-OA and dentin-pulp injury. Besides, the ratio of OPG/RANKL plays a regulatory role in OITRR and OTM, which suggests that boosting the amount of OPG may be a novel treatment idea. Nevertheless, the practical application in the human body is limited and its mechanism of is still in the exploratory stage. In addition, LIPUS mainly serves as a supporting measure in most oral diseases, which can speed up the treatment process of the disease, shorten the treatment time, and alleviate the pain. LIPUS is suggested for application with various parameters in treating different diseases. We found that the most common parameters of LIPUS are 1.5 MHz and 30 maw/cm$^2$ for 20 min/day (Table 1).

However, we found that the signaling pathways activated by LIPUS may also participate in the development of multiple diseases, such as organ fibrosis [128]. Therefore, whether the application of LIPUS in the human body will promote the occurrence and development of fibrosis or result in the emergence of adverse side effects is worth considering. In this case, the maximum single-use time and the maximum frequency of LIPUS should also be carefully studied. The lack of clarity on the use of LIPUS may also be one of the reasons why LIPUS has shown limited studies in humans. Actually, many effects of LIPUS remain unknown, so it is still necessary to strengthen the research on LIPUS in the future, especially regarding the underlying cellular and molecular mechanisms. It would be helpful to design models that can better simulate the human environment to explore the mechanisms of action of LIPUS in different oral diseases or the relationship between different mechanisms to make it more targeted to treat diseases. Besides, there is less research on implant osseointegration and dentin-pulp complex, so more research is required to expand the understanding of LIPUS. However, in studies on the bone biological role of LIPUS, animal experiments constitute the majority and animal models are mostly based on the tibia rather than the jaw. Further studies are needed to verify whether there are differences in LIPUS used in the tibia and jaw. More human trials or human tooth slice organ experiments are also urgent to validate the efficacy and safety of LIPUS. More importantly, the optimal frequency and duration of LIPUS in treating different oral diseases need to be found through controlled experiments in order to enhance the effectiveness and efficiency of the application of LIPUS.

**Table 1.** Targets, parameters, and effects of LIPUS in oral diseases.

| Field of Employment | Targets | The Parameter of LIPUS | Effect | Reference |
|---|---|---|---|---|
| Periodontal tissue regeneration | osteoblasts, cells in periodontal ligament and gingival epithelium | Pulse frequency: 1.5 MHz<br>Intensity: 30 mW/cm$^2$<br>Time: 20 min/day for four weeks | periodontal wound healing and bone repair | Ikai, Tamura [129] |
| | ECM protein in hPDLSCs | Pulse frequency: 1.5 MHz<br>Intensity: 90 mW/cm$^2$<br>Time: 30 min/day in a 37 °C water bath | Promote the osteogenic differentiation of hPDLSCs and induce the regeneration of periodontal ligament | Li, Zhou [14] |
| | unfolded protein reaction (UPR) in PDLSCs | Pulse frequency: 1.5 MHz<br>Intensity: 90 mW/cm$^2$<br>Time: 30 min/day | Enhance the osteogenic ability of PDLSCs and reduce the inflammatory response | Li, Deng [17] |
| | NF-κB pathway | Pulse frequency: 1.5 MHz<br>Intensity: 30, 60, 90 mM/cm$^2$<br>Time: 15 min/day for 7 days | Facilitate the immunoregulation and osteogenic ability of hPDLSCs | Lin, Wang [26] |
| | TWIST1/SDF-1 signaling pathway | Intensity: 90 mW/cm$^2$<br>Time: 30 min/day | Promote PDLSCs migration | Wang, Li [29] |
| | p38 MAPK pathway | Pulse frequency: 1 MHz<br>Intensity: 90 mW/cm$^2$<br>Time: 0 min, 15 min, 30 min, 60 min, 90 min, 120 min, 6 h | Contribute to PDLCs osteogenic differentiation | Ren, Yang [34] |
| | bone morphogenetic protein-smad signaling | Pulse frequency: 1.5 MHz<br>Intensity: 90 mW/cm$^2$<br>Time: 20 min/day | Accelerate PDLCs osteogenic differentiation | Yang, Ren [36] |
| | miR-182 | Pulse frequency: 1.5 MHz<br>Intensity: 90 mW/cm$^2$ | Enhance PDLCs osteogenic differentiation | Chen, Xiang [39] |
| | HGF | Pulse frequency: 1.5 MHz<br>Intensity: 30 mW/cm$^2$<br>Time: 5 min/day or 10 min/day for 1~4 weeks | Promote HGF differentiation | Mostafa, Uludağ [43] |
| OTM | HGF/Runx2/BMP-2 signaling pathway | Pulse frequency: 1.5 MHz<br>Intensity: 30 mW/cm$^2$<br>Time: 20 min/day | Accelerate OTM and alveolar bone remodeling | Xue, Zheng [59] |
| OITRR | OPG, RANKL, Cox-2 | Pulse frequency: 1.5% ± 5% MHz<br>Intensity: 30% ± 30% mW/cm$^2$ | OITRR inhibition and repair | Gul Amuk, Kurt [68] |
| Dental implant | local neuronal | Pulse frequency: 1 MHz<br>Intensity: 30 mW/cm$^2$<br>Time: 20 min/day for 14 or 28 days | Facilitate peri-implant osseointegration | Jiang, Yuan [83] |
| TMJ-OA | Sox9, Collagen II, Aggrecan, VEGF | Pulse frequency: 1 MHz<br>Intensity: 45 mW/cm$^2$<br>Time: 20 min/day for 3 days | Promote the recovery of injury chondrocytes | Du, Liang [97] |
| | HIF pathway | Pulse frequency: 1 MHz<br>Intensity: 45 mW/cm$^2$<br>Time: 20 min/day for 4 weeks | Reduce chondrocytes injury | Yang, Liang [98] |
| | ZNT-9 | Pulse frequency: 1 MHz<br>Intensity: 100 mW/cm$^2$<br>Time: 20 min/day for 5 days a week | Protect chondrocytes | He, Wang [99] |
| | TGF-β1/Smad3 pathway | Pulse frequency: 1 MHz<br>Intensity: 30 mW/cm$^2$<br>Time: 20 min/day for 6 weeks | Reduce inflammatory response | Yi, Liu [101] |
| Mandible fracture | BMP | Pulse frequency: 1.5 MHz<br>Intensity: 30 mW/cm$^2$<br>Time: 20 min/day for 4, 8, 14, 20 days | Promote bone repair | Huang, Hasegawa [115] |
| Dentin-pulp injury | TGF-β1 and Smad 2, 3 | Pulse frequency: 1.5 MHz<br>Intensity: 30 mW/cm$^2$<br>Time: 20 min/day for 1, 3, 5, 7, 14 days | Dentin-pulp-repair after injury | Wang, Zuo [122] |
| | calcium transport-related proteins | Pulse frequency: 1.5 MHz<br>Intensity: 30 mW/cm$^2$<br>Time: 20 min/day for 1, 3, 7, 14 days | Accelerate the formation of tertiary dentin | Zuo, Zhen [123] |

In conclusion, LIPUS was proven to have broad application prospects in the prevention and treatment of oral diseases, but its application remains a great challenge. Because LIPUS has a positive effect on a wide range of oral diseases, it will be possible to simultaneously

alleviate a variety of oral disorders by irradiating LIPUS. Through further research on LIPUS, it is possible that LIPUS will become an important tool for oral disease treatment and be more widely used in the future.

**Author Contributions:** Y.W. wrote the manuscript and drew the pictures. Y.G. conceived the study and revised the manuscript finally. All authors have read and agreed to the published version of the manuscript.

**Funding:** This work was supported by the National Natural Science Foundation of China (No. 32171308), the Technological Innovation R&D Project of Chengdu (No. 2021-YF05-02097-SN), and the Science and Technology Project of Sichuan Province (No. 2021YJ0150).

**Conflicts of Interest:** There is no conflict of interest for all the authors.

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
