# Peer review of "Clinical Applications of Low-Intensity Pulsed Ultrasound and Its Underlying Mechanisms in Dentistry"

_applsci, doi:10.3390/app122311898_

Round 1
Reviewer 1 Report
Very interesting and well written article.
Author Response
Dear Reviewer,
Thank you for your comments. Please see the attachment for our responses.
With best regards,
Yuzi Wei, Yongwen Guo

Reviewer 2 Report
Dear Authors,
I’ve extensively read the manuscript titled “Clinical applications of low-intensity pulsed 2 ultrasound and its underlying mechanisms in 3 dentistry”. The aim of this study is to elucidate the Low-intensity pulsed ultrasound (LIPUS) technology and how it serves as a non invasive treatment tool that reaches the lesion site in the form of ultrasound. The methodology is appropriate and quite linear with recent evidences/ studies on this topic.
Some aspects must be improved.
1. In the abstract, authors must clearly state and the aim of the paper, what the reader should be expected to find reading this study.
2. In the introduction, authors must clearly state and the aim of the paper, what the reader should be expected to find reading this study. They must state the novelty of their study, for example the completeness of their review on the topic addressed.
3. a significant revision of scientific English is warmly recommended
4. authors should improve and better argue about the orthodontic application of LIPUS, in particular in comparison with photobiomodulation or low-level laser therapy, reporting appropriate scientific evidence
Caccianiga G, Crestale C, Cozzani M, Piras A, Mutinelli S, Lo Giudice A, Cordasco G. Low-level laser therapy and invisible removal aligners. J Biol Regul Homeost Agents. 2016; 30 (2 Suppl 1:107-13.
Lo Giudice A, Nucera R, Leonardi R, Paiusco A, Baldoni M, Caccianiga G.
Another orthodontic application is the effect of orthodontic forces on dental pulp. Authors should introduce this concept and potential application of LIPUS in the orthodontic section on in the dental pulp paragraph
Lo Giudice A, Leonardi R, Ronsivalle V, Allegrini S, Lagravère M, Marzo G, Isola G. Evaluation of pulp cavity/chamber changes after tooth-borne and bone-borne rapid maxillary expansion. A CBCT study using surface-based superimposition and deviation analysis. Clin Oral Inv. 2020; 25 (4): 2237-2247
5. authors should argue about the potential application of LIPUS in influencing the synovial tissue in subjected affected by temporomandibular disorders, reporting appropriate scientific evidence
Loreto C, Filetti V, Almeida LE, La Rosa GRM, Leonardi R, Grippaudo C, Lo Giudice A. MMP-7 and MMP-9 are overexpressed in the synovial tissue from severe temporomandibular joint dysfunction. Eur J Histochem. 2020 Apr 16;64(2): 3113
Lo Giudice A, Rustico L, Caprioglio A, Migliorati M, Nucera R. Evaluation of condylar cortical bone thickness in patient groups with different vertical facial dimensions using cone-beam computed tomography. Odontology 2020;108(4):669-675
Author Response

(The authors gave the same response as above.)

Reviewer 3 Report
While the subject matter of the manuscript is very interesting, the article overall is too long. Please abbreviate each section without breaking the semantic integrity.
Abstract: There is no semantic integrity in the sentence that starts on line 26 and ends on line 31. Please rephrase the sentence in a more meaningful way.
In the sentence that starts on line 31 and ends on line 35, the sentence should be rewritten in a passive language in an official language.
Table: Present the references in another format in the reference column in Table 1. For example, a format containing author names may be preferred.
Conclusion: It is stated in this manuscript that the studies conducted are mostly in vitro studies and that in vivo studies are very limited. However, further information on further limitations of these studies should be given in this section.
Figures: The figures have been studied in detail. An important detail for understanding the assembly. Thanks to the authors for that.
Author Response

(The authors gave the same response as above.)

Reviewer 4 Report
1.The Authors used a lot of abbreviations without explanation. Please explain the abbreviation when first mentioned.
FAK line 23
PDLSCS, PCR e.c.t.
Abstract
2. The conclusions do not correspond to the aim of the study.
3. What was the methodology? How were articles classified for the manusccript?
Introduction
52-”It often adopts frequencies of 1~3 MHz and52
the length of this therapy lasts 5~20min every day.”
In all cases? The authors should add what the dose and time of use depend on.
129- mandibular osteoblasts- what do you mean?
139-“However, a number of studies” -references
148- PDLSCs- abbreviation
157 PCR
322”-In most of the patients” In what percentage of cases?
634- ”Dental-pulp complex consists of dentin and dental pulp and
this complex plays a protective role.”
Only protective? Authors should add other roles for the complex.
652- “drilling a hole” ??
This is not a medical language.
Author Response

(The authors gave the same response as above.)

Round 2
Reviewer 2 Report
the authors have successfully satisfied my previous concerns. the manuscript can be published
Reviewer 3 Report
Thanks to the authors for the corrections. As such, it is acceptable.
Reviewer 4 Report
The authors responded to all comments in the review. I have no more comments.